



# Stratospheric water vapor and ozone response to different QBO disruption events in 2016 and 2020

Mohamadou A. Diallo[1], Felix Ploeger[1, 2], Michaela I. Hegglin[1, 2, 3], Manfred Ern[1], Jens-Uwe Grooß[1], Sergey Khaykin[4], and Martin Riese[1, 2]

[1]Institute of Energy and Climate Research, Stratosphere (IEK–7), Forschungszentrum Jülich, 52 425 Jülich, Germany.
[2]Institute for Atmospheric and Environmental Research, University of Wuppertal, Wuppertal, Germany.
[3]Department of Meteorology, University of Reading, Reading, UK.
[4]Laboratoire Atmosphères, Milieux, Observations Spatiales, UMR CNRS 8190, IPSL, Sorbonne Univ./UVSQ, Guyancourt, France.

**Correspondence:** Mohamadou A. Diallo (m.diallo@fz-juelich.de)

**Abstract.** The Quasi-Biennial Oscillation (QBO) is a major mode of climate variability with periodically descending westerly and easterly winds in the tropical stratosphere, modulating transport and distributions of key greenhouse gases such as water vapor and ozone. In 2016 and 2020, anomalous QBO easterlies disrupted the QBO's 28–month period previously observed. Here, we quantify the impact of these two QBO disruption events on the Brewer–Dobson circulation, water vapour and ozone using the ERA5 reanalysis and satellite observations, respectively. Both lower stratospheric trace gases decrease globally

during the 2015–2016 QBO disruption event, while they only weakly decrease during the 2019–2020 QBO disruption event. These dissimilarities in the circulation anomalous response to the QBO disruption events result from differences in the tropical upwelling caused by anomalous planetary and gravity wave forcing in the lower stratosphere near the equatorward flanks of the subtropical jet. The differences in the response of lower stratospheric water vapor to the 2015–2016 and 2019–2020

QBO disruption events are due to the cold–point temperature differences induced by the Australian wildfire, which moistened the lower stratosphere, therefore, hidding the 2019–2020 QBO disruption impact. Our results highlight the need for a better understanding of the causes of QBO disruption events, their interplay with other climate variability modes, and their impacts on water vapor and ozone in the face of a changing climate.

# 1 Introduction

The upper troposphere and lower stratosphere (UTLS) is a key region of the Earth climate system because of its large sensitivity to radiative forcing of greenhouse gases, such as water vapor ($H_2O$) and ozone ($O_3$) (Gettelman et al., 2011; Dessler et al., 2013; Nowack et al., 2015). Any changes in the composition of these radiatively active trace gases in the UTLS region induced by the stratospheric Brewer-Dobson circulation (BDC) and its modulation by the modes of climate variability lead to large impact



on surface climate (e.g., Forster and Shine, 2002, 1999; Solomon et al., 2010; Riese et al., 2012; Butchart, 2014; Diallo et al.,
     2017, 2018, 2019, 2021). Ozone is mainly produced in the middle stratosphere and is a good proxy of the tropical upwelling.
     In addition, ozone variability in the tropical lower stratosphere is affected by variability in tropical upwelling of the BDC
     (Randel et al., 2007; Abalos et al., 2013; Stolarski et al., 2014). The ozone transport and lifetime in the UTLS region are both
     modulated by the seasonality in the BDC and the natural modes of climate variability, including the Quasi-Biennial Oscillation

(QBO) (Randel and Thompson, 2011; Diallo et al., 2018). Lower stratospheric water vapor and its multi-timescale variations
     ranging from day to decades are mainly controlled by changes in the tropical cold point temperatures and its modulations by
     the natural modes of climate variability, including the QBO (Holton and Gettelman, 2001; Hu et al., 2016; Diallo et al., 2018;
     Tao et al., 2019; Randel and Park, 2019). Therefore, the amount of water vapor in the UTLS region is directly linked to the
     dehydration in the air parcels crossing through the coldest temperatures in the tropical tropopause layer (e.g., between 14 and

19 km; Fueglistaler et al., 2009).

     Considered as a dominant mode of variability of the equatorial stratosphere, the QBO globally impacts the transport and
     distributions of stratospheric water vapor and ozone. Mostly driven by gravity waves and equatorially trapped waves, the QBO
     is a quasi-periodic oscillation between tropical westerly and easterly zonal wind shears (Baldwin et al., 2001; Ern et al., 2014).
     Both QBO phases modulate the vertical and meridional components of the BDC and affect temperature structure, therefore,

impacting the water vapor and ozone composition and radiative feedback in the UTLS region (Plumb and Bell, 1982; Niwano
     et al., 2003; Diallo et al., 2018).

     The quasi-periodic QBO cycle of about 28–month period, which alternates between westerly and easterly zonal wind shears,
     was subject to two disruptions in the past five years. In January 2016 and 2020, the anomalous QBO westerlies in the tropical
     lower stratosphere were unexpectedly interrupted by anomalous QBO easterlies caused by planetary waves propagating from

the mid-latitudes toward the equatorial region combined with equatorial convective gravity waves (Osprey et al., 2016; Coy
     et al., 2017; Kang et al., 2020; Kang and Chun, 2021). There is not yet a clear understanding of how these QBO disruptions are
     linked to anomalously warm or cold sea surface temperatures (Taguchi, 2010; Schirber, 2015; Dunkerton, 2016; Christiansen
     et al., 2016; Barton and McCormack, 2017), volcanic aerosols (Kroll et al., 2020; DallaSanta et al., 2021), wildfire smoke
     (Khaykin et al., 2020; Yu et al., 2021; Peterson et al., 2021) and climate changes (Anstey et al., 2021b). However, recent study

based on climate model simulations from phase six of the Coupled Model Intercomparison Project (CMIP6) predicts increased
     disruption frequencies to the quasi-regular QBO cycle in a changing climate (Osprey et al., 2016; Anstey et al., 2021b).
     Previous studies also suggest that the QBO amplitude in the tropical stratosphere is decreasing in the lower stratosphere due
     to the climate change–induced strengthening of tropical upwelling (Saravanan, 1990; Kawatani et al., 2011; Kawatani and
     Hamilton, 2013). Thus, in the context of a changing climate, the predictable QBO signal associated with the quasi-regular

phase progression and amplitude as well as its potential impacts on UTLS composition faces an uncertain future. Therefore, it
     is of particular importance to quantify and better understand the different impact of the QBO disruption events on UTLS water
     vapor and ozone, which have the potential to globally affect the radiative forcing of the Earth's climate system (Forster and
     Shine, 1999; Butchart and Scaife, 2001; Solomon et al., 2010; Riese et al., 2012).





Here, we quantify the similarity and differences in the strength and depth between the 2015–2016 and 2019–2020 disrupted
QBO impacts on lower stratospheric water vapor and ozone from satellite observations. Also, we analyse the main drivers of
the differences in anomalous circulation and UTLS composition changes. Section 2 describes the satellite observational data
sets and the multivariate hybrid regression model used for the quantification. Section 3 describes the anomalous BDC and
UTLS composition changes following the 2016 and 2020 QBO disruption events. Section 4 discusses the results of a well-
established multivariate hybrid regression analysis to provide evidence for the impact of the QBO disruption events on lower
stratospheric water vapor and ozone. Finally, we discuss the main reasons of the anomalous BDC and UTLS composition
differences between the 2015–2016 and 2019–2020 disrupted QBO impacts in relationship to planetary and gravity wave
dissipation likely caused by the anomalous surface conditions associated with the strong El Niño Southern Oscillation (ENSO)
in 2015–2016, the strong Indian Ocean Dipole (IOD) in 2019–2020. We further discuss the differences between 2016 and 2020
in view of the particularly warmer stratosphere linked to Australian wildfire smoke in 2020.

## 2   Data and methodology

To quantify the QBO impact, we used the monthly mean zonal mean ozone and water vapor mixing ratios from the Aura
Microwave Limb Sounder (MLS) satellite observations covering the 2005–2020 period (Livesey et al., 2017). The version 4.4
MLS data set used here has a vertical resolution of 2.5–3 km ranging from 8 to 35 km and 60 °S/N with a high precision and
lower systematic uncertainty (Santee et al., 2017). Previous findings show that MLS zonal monthly mean $H_2O$ mixing ratios
show very good agreement with the multi-instrument mean (Hegglin et al., 2013, 2021).

In addition to the MLS observation data sets, we also utilize the temperature ($T$) and zonal mean wind ($U$) from the ERA5
reanalysis of the European Centre for Medium-Range Weather Forecasts (ECMWF) (Hersbach et al., 2020). We have also
calculated the residual circulation vertical velocity ($\overline{w^*}$) using the Transformed Eulerian Mean (TEM; Andrews et al. (1987))
and decomposed the wave drag into planetary (PWD) and gravity (GWD) wave drag contributions to the circulation anomalies
(Ern et al., 2014, 2021). Note that we are using the ERA5 reanalysis data on the original 137 model levels for calculating the
TEM budget, but not the coarse conventional pressure-level data, which can cause large uncertainties in the equatorial waves
and zonal wind in the tropical stratosphere (Fujiwara et al., 2012; Kim and Chun, 2015; Kawatani et al., 2016). For more details
about the ERA5 TEM calculations and wave decomposition please see Diallo et al. (2021).

We disentangle the QBO impact on these monthly mean zonal mean stratospheric water vapor and ozone mixing ratios from
the other sources of natural climate variability using a multivariate hybrid regression model for the 2005–2020 period (Eq. 1).
To highlight the two QBO disruptions, figures only show the 2013–2020 period. The established multivariate hybrid regression
method is appropriate to separate the relative influences of the considered modes of climate variability, including the QBO, on
stratospheric water vapor and ozone. Additional details about the multvariate hybrid regression model and its applications can
be found in Diallo et al. (2018). Our multvariate hybrid regression model decomposes the given monthly zonal mean variable,
$Var_i$, into a long-term linear trend, seasonal cycle, modes of climate variability and a residual ($\varepsilon$). For a given variable $Var_i$
(herein $H_2O$, $O_3$, $\overline{w^*}$, T, PWD and GWD), the multivariate hybrid regression model yields



$$Var_i(t_{month}, y_{lat}, z_{alt}) = Trend(t_{month}, y_{lat}, z_{alt}) + SeasCyc(t_{month}, y_{lat}, z_{alt}) + \sum_{n=1}^{5} b_n(y_{lat}, z_{alt}) \cdot Proxy_n(t_{month} - \tau_n(y_{lat}, z_{alt})) +$$

$$\varepsilon(t_{month}, y_{lat}, z_{alt}), \tag{1}$$

where $Proxy_n$ represents the different climate indexes used here. $Proxy_1$ is a normalized QBO index (QBOi) from the tropical ERA5 zonally averaged tropical zonal mean winds with full vertical levels then deseasonalised and normalized by the standard deviation to build the QBOi (Hersbach et al., 2020). $Proxy_2$ is the normalized Multivariate ENSO Index (MEI; Wolter and Timlin, 2011), $Proxy_3$ is the Indian Ocean Dipole (IOD, Saji et al., 1999), $Proxy_4$ is the Madden-Julian Oscillation (MJO, Son et al., 2017), and $Proxy_5$ is the AOD from satellite data (Thomason et al., 2018). $Trend(t_{month}, y_{lat}, z_{alt})$ is a linear trend. $SeasCyc(t_{month}, y_{lat}, z_{alt})$ is the annual cycle. The coefficients are the amplitude $b_n$ and the lag $\tau_n(y_{lat}, z_{alt})$ associated with the QBO, ENSO, IOD, MJO and AOD respectively. The solar forcing is neglected because our data set is relatively short. Finally, we estimate the uncertainty in the multivariate hybrid regression model using a Student's $t$ test technique (von Storch and Zwiers, 1999; Friston et al., 2007).

## 3 Characterisation of the 2016 and 2020 anomalous circulations

In In February 2016 and January 2020 unexpected tropical QBO easterlies (negative QBOi) developed in the center of the tropical QBO westerlies, thereby breaking the quasi-regular QBO cycle of alternating easterly and westerly phases (Fig. 1a) (Osprey et al., 2016; Newman et al., 2016; Anstey et al., 2021a). Both QBO disruption events have been associated with a combination of extratropical Rossby waves, equatorial planetary waves (Kelvin, Rossby, mixed Rossby–gravity, and inertia–gravity), and small-scale convective gravity waves, propagating into the deep tropics and depositing their negative momentum forcing (Osprey et al., 2016; Newman et al., 2016; Kang et al., 2020; Kang and Chun, 2021). Both QBO disruption events are primarily triggered by mid-latitude Rossby waves propagating from the northern hemisphere in 2015–2016 and from the southern hemisphere in 2019–2020 into the deep tropics. In 2015–2016, the equatorial planetary wave forcing may have preconditioned mid-latitude Rossby waves to break easily at the equator (e.g. Lin et al. (2019)), while in 2019–2020, the equatorial planetary and small-scale convective gravity waves propagating into the UTLS predominantly contributed to the disruption (Kang et al., 2020; Kang and Chun, 2021). Note that the potential processes and mechanisms triggering the QBO disruption are still under debate. Recent findings from Match and Fueglistaler (2021) using a 1D theorical model of the QBO from Plumb and Bell (1982) pointed out the key role of the upwelling and wave dissipation. For more details about the triggering please see these studies (Schirber, 2015; Dunkerton, 2016; Christiansen et al., 2016; Coy et al., 2017; Barton and McCormack, 2017; Hitchcock et al., 2018; Watanabe et al., 2018; Renaud et al., 2019; Match and Fueglistaler, 2021). Although similar in many respects, including the causes of the sudden development of tropical QBO easterlies in the center of tropical QBO westerlies, the two disruptions also exhibit differences, particularly in the structure (strength and depth) of the impacts and the level at which it started. Here, we mainly focus on the impact of the QBO disruption events on the lower stratospheric BDC and trace gases, including water vapour and ozone.



**Figure 1.** Tropical average of the zonal mean zonal wind (U) from ERA5 **(a)** and deseasonalized stratospheric $H_2O$ and $O_3$ time series from MLS satellite observations for the 2013–2020 period in percent change from long-term monthly means as a function of time and altitude. Shown are **(a)** Zonal mean zonal wind U, **(b)** Deseasonalized monthly mean $H_2O$ anomalies, **(c)** Deseasonalized monthly mean $O_3$ anomalies. Vertical grey dashed lines indicate the QBO disruption onset and offset years. The lowermost panel **(d)** shows the QBO index at $50\,hPa$ in red, the MEI index in blue and the AOD index in black. Monthly averaged zonal mean zonal wind component, $u$ ($\mathrm{m\,s^{-1}}$), from ERA5, is overlaid as solid white (westerly wind) and dashed gray (easterly wind) contour lines.


The similarities as well as the differences between the two disruption events are also visible in the inter-annual variability of the tropical lower stratospheric zonal mean zonal wind (a), $H_2O$ (b) and $O_3$ (c) anomalies as a percentage change relative to the

120 monthly mean mixing ratio during the 2013–2020 period (Fig. 1a–c). Both QBO disruption events are expected to impact the tropical upwelling of the BDC through the two way interactions between the mean–flow and wave propagation associated with the QBO phases (Plumb, 1977; Lindzen, 1971; Holton, 1979; Dunkerton, 1980; Plumb and Bell, 1982; Grimshaw, 1984; Match and Fueglistaler, 2021) as well as through its control of the tropical cold point temperatures (Kim and Son, 2012; Kim and Chun, 2015). This impact of the 2015–2016 and 2019–2020 QBO disruption events on the transport and distribution of lower

stratospheric $H_2O$ and $O_3$ mixing ratios is the most effective when the signal reaches the tropical cold point temperature altitude ($\sim 17\,km$) e.g. from June to December in 2016 and from June to August in 2020 (Fig. 1) (Tweedy et al., 2017; Diallo et al., 2018). The zonal mean zonal wind shows that the westerly jet at $30\,hPa$ is stronger and deeper during the 2015–2016 QBO disruption than the 2019–2020 QBO disruption (Fig. 1a and Fig. S1a–b in the supplement). The 2019–2020 QBO disruption shows a clear cut of the westerlies into two parts while the 2015–2016 QBO disruption shifts the westerlies upward (Fig. 1a).

As soon as the downward propagation of tropical QBO easterlies reaches the tropical cold point temperature ($\sim 17\,km$) from June to December 2016, the $H_2O$ mixing ratios decrease i.e. turning from positive to negative anomalies. As reported by Diallo et al. (2018), the alignment of the strong El Niño event with westerly QBO in early boreal winter of 2015–2016 (September 2015–March 2016) substantially increased $H_2O$ mixing ratios and decreased $O_3$ mixing ratios up to about 20 % in the tropical lower stratosphere between the tropopause and the altitude of $25\,km$ (Fig. 1b, c). Then, the sudden occurrence of the QBO

disruption decreased the lower stratospheric $H_2O$ and $O_3$ mixing ratios from late spring to early following winter up to about 20 %.

Conversely during the 2019–2020 QBO disruption, Figure 1b, c show clear differences in the tropical lower stratospheric trace gas anomalies, particularly in the strength and depth of $H_2O$ and $O_3$ anomalies, consistent with the strutural zonal mean zonal wind changes (Fig. S1a, b). The tropical lower stratospheric $O_3$ anomalies are purely responding to the enhanced tropical

upwelling of the BDC caused in 2016 by a combination of a strong El Niño event, negative IOD event and the QBO disruption in 2015–2016, and in 2020 by a combination of a weak La Niña, strong positive IOD event and the QBO disruption in 2019–2020 (e.g., easterly winds at $100–40\,hPa$). Tropical lower stratospheric $O_3$ anomaly is a good proxy of the tropical upwelling of the BDC as its concentration is modulated by the advection of tropospheric air generally poor in $O_3$ into the stratosphere (Randel et al., 2006; Abalos et al., 2013; Stolarski et al., 2014; Weber et al., 2011; Iglesias-Suarez et al., 2021). The small

decrease in the tropical lower stratospheric $O_3$ anomalies up to about 10 % in 2020 compare to about 20% in 2016 between the altitude of $16\,km$ and $25\,km$ suggests a stronger tropical upwelling and its modulations in 2016 than in 2020 (Fig. 1c and Fig. S3a in the supplement).

The tropical lower stratospheric $H_2O$ variability (tape recorder) is more challenging to interpret because of its regulation by the variability in the tropical cold point temperatures (Holton and Gettelman, 2001; Hu et al., 2016; Randel and Park,

2019). The negative tropical lower stratospheric $H_2O$ anomalies induced by the interplay of different modes of natural climate variability, including the QBO, are weaker in 2020 than in 2016 (Fig. 1b and Fig. S2a, b in the supplement). The tropical lower stratospheric $H_2O$ anomalies averaged between the altitude of $16\,km$ and $20\,km$ are up to about 20 % more negative in





2016 than in 2020 (Fig. S3a in the supplement). Particularly, the 2020 tape recorder shows large positive $H_2O$ anomalies even after the QBO disruption that are of opposite sign to the 2016 $H_2O$ anomalies (Fig. 1b). This complexity in $H_2O$ inter-annual

variability lies in its dependency on the interplay of different modes of natural climate variability, including the QBO phases (Diallo et al., 2018; Brinkop et al., 2016; Tian et al., 2019; Liess and Geller, 2012), seasons (early or late in the winter) and location (western, central or eastern Pacific, where the ENSO and IOD maximum occurs (Garfinkel et al., 2013; Smith et al., 2021)). Therefore, to elucidate the effect of both QBO disruption events on the lower stratospheric $H_2O$ and $3_3$ anomalies, we performed regression analysis both without and with explicitly including QBO signals to isolate the QBO impact on these

trace gases. The difference between the residual ($\varepsilon$ in Eq. 1) with and without explicit inclusion of the QBO signals gives the QBO–induced impact on stratospheric $H_2O$ and $O_3$ anomalies. This approach of differencing the residuals is similar to direct calculations, projecting the regression fits onto the QBO basis functions, i.e., the QBO predictor timeseries (see supplement Figs. 2 and 4 in (Diallo et al., 2017)). In addition, this differencing approach avoids the need to reconstruct the time series after the regression analysis.

## 4    Drivers detection and attribution of the anomalous circulations

### 4.1    Impact of QBO disruptions on UTLS composition

Figures 2a, b show time series of the QBO–induced inter-annual variability in tropical lower stratospheric $H_2O$ and $O_3$ anomalies estimated from the difference between the residual ($\varepsilon$ in Eq. 1) without and with explicit inclusion of the QBO proxy for the 2013–2020 period. A footprint of both QBO disruption events is clearly visible in lower stratospheric $H_2O$ and $O_3$ anoma-

lies with a shift from positive anomalies related to the westerly winds (positive QBOi) to negative anomalies related to the easterly winds (negative QBOi). The QBO disruption–induced $O_3$ anomalies are sudden and clearly follow the monthly mean zonal mean wind changes. The QBO disruption–induced $H_2O$ anomalies are roughly in phase with the zonal wind anomalies with a delay of about 3–6 months because of the $H_2O$ tropospheric origin as well as its dependency on the tropical cold point temperature anomaly.

Beside the good agreement in the structure of both trace gas changes, there are clear differences in the strength and depth of both lower stratospheric $H_2O$ and $O_3$ responses to the QBO disruptions between the 2016 and the 2020 events and, particularly large for the $H_2O$ response. These differences in the impact of the QBO disruption events are consistent with the observed lower stratospheric $H_2O$ and $O_3$ anomalies (Fig. 1, Fig. S2 and S3). During 2016, the QBO shift from westerlies to easterlies at $40\,hPa$ in the tropical lower stratosphere induces substantial negative $H_2O$ and $O_3$ anomalies up to about $20\,\%$ between the

altitude of $16\,km$ and $25\,km$ from the early boreal spring to the next boreal winter (Fig. 2). This decrease in $H_2O$ and $O_3$ mixing ratios is consistent with upward transport of young and dehydrated air poor in $H_2O$ and $O_3$ into the lower stratosphere between the altitude of $16\,km$ and $25\,km$. As expected, the sudden occurrence of the QBO disruption events caused anomalously cold point temperatures and enhanced tropical upwelling in 2016 and in 2020, consistent with the decrease in $H_2O$ and $O_3$ mixing ratios indueced by the QBO easterly (Fig. 2). However, besides the similarities in the structural changes, the negative $H_2O$

and $O_3$ anomalies induced by the QBO disruption are smaller and shallower in 2020 than in 2016. While the differences in





the $O_3$ anomalies induced by the QBO disrution events are small between the year 2016 and year 2020, the differences in the disrupted QBO impact on $O_3$ mixing ratios are particularly large between the year 2016 and year 2020 (Fig. 2a and Fig. S3b in the supplement). The differences in the magnitude of negative $O_3$ anomalies suggest a slightly weaker modulation of the anomalous tropical upwelling of the BDC by the secondary circulation in 2020 than in 2016, consistent with the differences

in the strength and depth of the residual vertical velocity and wave forcing anomalies discussed in Sect. 4.2. The differences in the strength and depth of the $H_2O$ response to the QBO disruption events suggest that the tropical cold point temperature is substantially different between the year 2016 and year 2020. In addition, we note that the early QBO westerly followed by the shift to QBO easterly is not the main cause of the large increase in the 2020 lower stratospheric $H_2O$ anomalies. In the following, we assess the potential impact of the unusually strong Australian wildfire smoke on the lower stratospheric $H_2O$

anomalies in 2020 through its impact on the stratospheric temperature anomaly (Khaykin et al., 2020; Yu et al., 2021; Peterson et al., 2021).

Figures 3a–d show the zonal mean impact of the QBO disruption events on lower stratospheric $H_2O$ and $O_3$ anomalies. Figure 3e shows the impact of 2020 Australian wildfire AOD on lower stratospheric $H_2O$ anomalies. The lower stratospheric $H_2O$ anomalies are averaged from July to December for 2016 and from July to September for 2020 respectively. We chose

different averaging periods for 2016 (July–to–December) and 2020 (July–August–September) to have similar zonal mean structure of the $H_2O$ and $O_3$ anomalies response to QBO disruptions, although their depth and strength are different from each other.

In 2016, the shift to QBO easterly phase in the tropics significantly dehydrates the global lower stratosphere up to about 20 % below the altitude of $20\,km$ (Fig. 3a and Fig. S2a) (Diallo et al., 2018; Tweedy et al., 2017). This decrease in $H_2O$ mixing ratios

is due to the enhanced tropical upwelling of the BDC, its modulation by the secondary circulation and the related decrease of tropical cold point temperature as discussed later in Sect. 4.2 (Diallo et al., 2018; Jensen et al., 1996; Hartmann et al., 2001; Geller et al., 2002; Schoeberl and Dessler, 2011). Because of the hemispheric asymmetry of the BDC strength, which is driven by planetary wave activity (e.g. Holton and Gettelman, 2001) and eddy mixing (e.g. Haynes and Shuckburgh, 2000), the rising dehydrated air from the tropics moves toward middle and high latitudes of both hemispheres, but stronger in winter

hemisphere. The positive $H_2O$ anomalies above the altitude of $20\,km$ are related to the effect of the preceding QBO westerly phase on tropical UTLS temperatures and the upward propagating tape-recorder signal. The changes in $H_2O$ anomalies are consistent with the observed negative tropical $O_3$ anomalies below the altitude of $20\,km$ induced by the QBO easterly phase (Fig. 3a, c and Fig. S2a, c in the supplement). These changes indicate an enhanced tropical upwelling of the BDC and its modulation by the QBO easterly phase in the lower stratosphere between the altitude of $16\,km$ and $20\,km$ (Fig. S4 in the

supplement). Above the altitude of $20\,km$, the positive tropical $O_3$ anomalies are associated with the QBO westerly phase (Fig. 3c and Fig. S2c in the supplement). Also note the large variability in extratropical $O_3$ anomalies related to the QBO influence on the extratropical circulation (Holton and Tan, 1980; Damadeo et al., 2014; Ray et al., 2020), stratospheric major warmings, and chemical processes (WMO, 2018).

In 2020, the QBO disruption–induced changes in tropical lower stratospheric $H_2O$ and $O_3$ anomalies exhibit similar struc-

ture to the effect of the 2015–2016 QBO disruption event. Note that we use different averaging periods for 2016 (July–to–

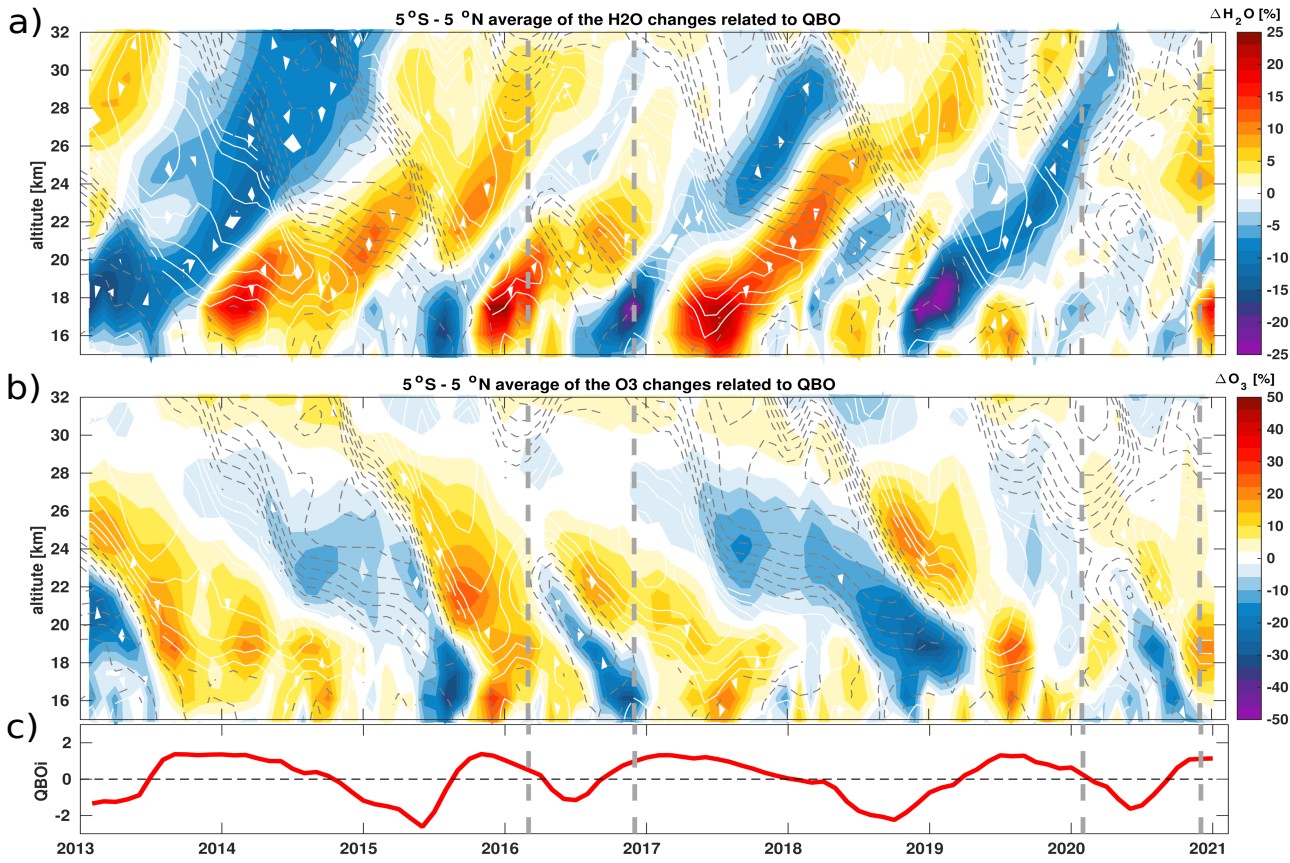

**Figure 2.** Tropical average of the QBO impact on the stratospheric $H_2O$ **(a)** and $O_3$ **(b)** anomalies from the MLS satellite observations for the 2013–2020 period in percent change relative to monthly mean mixing ratios as a function of time and altitude. Shown QBO impact on the stratospheric trace gases is derived from the multiple regression fit as the difference between the residual ($\varepsilon$ in Eq. 1) without and with explicit inclusion of the QBO signal. The lower panel below indicates the QBO index at $50\,hPa$ in red. Vertical grey dashed lines indicate the QBO disruption onset and offset years. Monthly averaged zonal mean zonal wind component, $u$ ($m\,s^{-1}$), from ERA5, is overlaid as solid white contours (westerly) and dashed gray contours (easterly) lines.

December) and 2020 (JAS) to highlight the structural similarities in the QBO impact. Both trace gases show negative anomalies in the tropics, corroborating the enhanced tropical upwelling of the BDC induced by the QBO shift from westerly winds to easterly winds in the tropics (Fig. 3b). However, there are also differences in both lower stratospheric $H_2O$ and $O_3$ responses to the shift from the tropical QBO westerly phase to the tropical QBO easterly phase between July–to–December 2016 and

JAS 2020. Note that the differences in the $H_2O$ response to the QBO disruption events between the year 2016 and the year 2020 are particularly large. Conversely to the globally dehydrated lower stratosphere in 2016, the sudden development of





**Figure 3.** Zonal mean impact of the QBO disruption on the lower stratospheric $H_2O$ **(a, b)** and $O_3$ **(c, d)** anomalies from MLS satellite observations averaged from July to December for 2016 **(a, c)** and from July to September for 2020 **(b, d)** period. In addition, the impact of the 2020 Australian wildfires is shown **(e)**. All panels show the percentage change relative to monthly mean mixing ratios as a function of latitude and altitude. The impact of the QBO disruptions and the Australian wildfire on the stratospheric trace gases is derived from the multiple regression fit as the difference between the residual ($\varepsilon$ in Eq. 1) without and with explicit inclusion of the QBO signal. The black dashed horizontal line indicates the tropopause from ERA5. Monthly averaged zonal mean zonal wind component, $u$ (m s$^{-1}$), from ERA5, is overlaid as solid white (westerly wind) and dashed gray (easterly wind) contours lines.





tropical QBO easterly in 2020 led to a small decrease in lower stratospheric $H_2O$ mixing ratios, therefore, to small negative lower stratospheric $H_2O$ anomalies (Fig. 3b). Despite the similar zonal mean structure of $O_3$ anomalies induced by both QBO disruption events within these different averaging periods for 2016 (July–to–December) and 2020 (JAS), the impact of the

QBO disruption on zonal mean $O_3$ mixing ratios are weaker when averaged in the entire year 2020 than in the year 2016 (Fig. 3c, d and Fig. S2c, d in the supplement). The differences in the strength and depth between the 2016 and 2020 $H_2O$ and $O_3$ anomalies and their modulation by the QBO disruption events clearly suggest substantial differences in the anomalous tropical upwelling of the BDC and the tropical cold point temperatures. The smaller negative tropical $O_3$ anomalies suggest that the tropical upwelling of the BDC and its modulation by the QBO–induced secondary circulation are weaker in 2020 than

in 2016. Simultaneously, the positive tropical $H_2O$ anomalies in 2020 that are not related to the QBO disruption (Fig. S2b) indicate a warmer tropical cold point temperature potentially induced by the unusually strong Australian wildfire smoke in the stratosphere (Khaykin et al., 2020; Yu et al., 2021; Peterson et al., 2021). The main dynamical causes of these differences are investigated in the following section.

## 4.2 Mechanisms driving the strength and depth differences

To further investigate and understand the key drivers of the anomalous circulation differences between the 2015–2016 and 2019–2020 QBO disruption events, we analyse the differences in the tropical upwelling of the BDC and the secondary circulation induced by the QBO wind shear. Figure 4a–d show time series of the tropical residual circulation vertical velocity and temperature anomalies together with the impacts of the two QBO disruption events on $\overline{w^*}$ and temperature anomalies during the 2015–2016 and 2019–2020 periods, respectively. Also, latitude–altitude sections of the $\overline{w^*}$ and temperatures together with the

associated impacts of the QBO disruption events during the 2015–2016 and 2019–2020 periods are shown in the supplement Fig. S4.

  Clearly, there are substantial differences in the anomalous tropical upwelling of the BDC as shown by the $\overline{w^*}$ and temperatures during the two disruption events, consistent with the $O_3$ anomalies (Fig. 1c). Also, the modulation of the tropical upwelling by the QBO exhibits differences but smaller than anomalous circulation differences, consistent with the QBO

disruption–induced $O_3$ anomalies (Fig. 2b). In 2016, the tropical upwelling anomalies strongly increases up to about 45 % below the altitude of about 20 $km$ from April to December when the QBO westerly phase shifts to QBO easterly phase (Fig. 4a). However in 2020, the tropical upwelling anomalies are weaker and only reach up to about 20 % below the altitude of about 20 $km$, leading to about 25 % weaker $\overline{w^*}$ anomalies in 2020 than in 2016 between the altitude of about 17 $km$ and 20 $km$. Below the altitude of about 17 $km$, the $\overline{w^*}$ anomalies are about 10 % weaker in 2020 than in 2016. In addition to the weaker tropical

upwelling in 2020, the impact of the QBO disruption events on $\overline{w^*}$ anomalies is consistent with the weaker QBO–induced secondary circulation in 2020 than in 2016 with up to about 25 % weaker modulation of the tropical upwelling (Fig. 4b). This weaker tropical upwelling of the BDC and the QBO–induced secondary circulation in 2020 than in 2016 is also visible in the zonal mean cross section of the mean $\overline{w^*}$ and temperature anomalies (Fig S4a, b, e, f in the supplement), together with the impacts of the QBO disruption events on $\overline{w^*}$ and temperature anomalies for 2016 and 2020 (Fig. S4c, d, g, h in the supplement).

The increase of the tropical upwelling as well as the secondary circulation induced by the QBO easterly wind shear are weaker



**Figure 4.** Tropical averaged of the deseasonalized mean residual vertical velocity ($\overline{w^*}$) **(a)** and temperature anomalies **(b)** time series from ERA5 for the 2013–2020 period together with the impact of QBO disruptions on the tropical mean $\overline{w^*}$ **(c)** and temperature anomalies **(d)** derived from the multiple regression fit as a function of latitude and altitude. **(a)** Deseasonalized monthly mean tropical upwelling. **(b)** Disrupted QBO impact on monthly mean tropical upwelling anomalies. **(c)** Deseasonalized monthly mean tropical temperature. **(d)** Disrupted QBO impact on monthly mean tropical temperature anomalies. Vertical grey dashed lines indicate the QBO disruption onset and offset years. The lowermost panel **(e)** shows the QBO index at $50\,hPa$ in red. Monthly averaged zonal mean zonal wind component, $u$ ($\mathrm{m\,s^{-1}}$), from ERA5, is overlaid as solid white (westerly) and dashed gray (easterly) contours lines.





and shallower in 2020 than in 2016 (Fig. 4a, b and Fig. S4a–d in the supplement). The differences in the anomalous tropical upwelling and secondary circulation are also consistent with the differences in the temperature anomalies as well as in the QBO disruption–induced temperature anomalies (Fig. 4c, d and Fig. S4e–h in the supplement). In 2016, the tropical cold point temperature anomalies (at altitudes of about 17–18 $km$) are substantially negative (Fig. 4c in the supplement). This decrease

in tropical temperatures is consistent with the strong tropical upwelling of the BDC and its modulation by the QBO–induced secondary circulation, which, in turn led to large negative tropical lower stratosphere $H_2O$ and $O_3$ anomalies in 2016 (Fig. 4 and Fig. S4a, c, e, g in the supplement).

Conversely, the tropical cold point temperature anomalies are warmer and barely exceeding -0.1 $K$ in 2020, consistent with the smaller tropical $\overline{w^*}$ anomalies (Fig. 4 and Fig. S4b, d, f, h in the supplement) and the shorter lifetime of tropical $O_3$

anomalies, which last for only about 3 months (Fig. 1 and Fig. 2). These warmer tropical cold point temperature anomalies corroborate the weaker tropical upwelling of the BDC and smaller tropical lower stratospheric $H_2O$ and $O_3$ mixing ratios in the year 2020. Interestingly, the differences in the tropical cold point temperature anomalies between the years 2016 and 2020 are more pronounced as shown in Figure S4e, f in the supplement than the differences in the QBO disruption–induced tropical cold point temperature anomalies (Figure S4g, h in the supplement). This anomalously warmer stratosphere, including warmer cold

point temperature in 2020, is consistent with recent findings about the impact of Australian wildfire smoke (Khaykin et al., 2020; Yu et al., 2021; Peterson et al., 2021). Therefore, we also pay attention to volcanic eruptions and Australian wildfire smoke in 2020, which can impact lower stratospheric temperatures, and therefore, lower stratospheric $H_2O$ and $O_3$ anomalies. Indeed using our regression analyses, we can show that the Australian wildfire largely moistened the lower stratosphere between the altitude of 16 $km$ and 25 $km$ in 2020 by inducing anomalously warmer stratosphere, thereby, hidding the impact of 2019–

2020 QBO disruption on $H_2O$ anomalies (Fig. 3e). The removal of Australian wildfire impact allows to better highlight the weak structure of the 2019–2020 disrupted QBO impact on lower stratospheric $H_2O$ anomalies, which is similar to the 2015–2016 QBO disruption-induced effect. Regarding the difference in the upwelling of the BDC forcing, we finally investigate the related wave drag changes in the following.

To investigate the main causes of the BDC differences between the year 2016 and the year 2020 during the QBO disruption

events, we calculate the planetary and gravity wave drag. We analyse the differences in terms of wave activities potentially induced by specific sea surface conditions such as the unsually warm 2015–2016 El Niño and the 2019–2020 strong positive Indian Dipole Ocean, which impact tropical convective activities (Jia et al., 2014). For additional details about the wave decomposition please see Diallo et al. (2021) and Ern et al. (2014).

The BDC and its interannual variability are driven by the planetary and gravity wave breaking in different stratospheric

regions (Haynes et al., 1991; Rosenlof and Holton, 1993; Newman and Nash, 2000; Plumb, 2002; Shepherd, 2007). Therefore, any changes in wave drag will lead to circulation and composition changes. Figure 5a–f show the January-to-June zonal mean of the deseasonalized monthly mean net wave forcing (PWD + GWD - du/dt), planetary wave drag (PWD) and gravity wave drag (GWD) from the ERA5 reanalysis for the years 2016 and 2020, respectively. Note that the net wave forcing (NetF) is equal to the contribution of Coriolis force plus meridional advection plus vertical advection to the momentum balance (Ern et al., 2021).

Clearly, the net forcing anomalies as well as the planetary and gravity wave drag anomalies exhibit differences in strength and





**Figure 5.** Deseasonalized monthly mean zonal mean net wave forcing (NetF)**(a, b)**, planetary wave drag (PWD) **(c, d)** and gravity wave drag (GWD) **(e, f)** anomalies from the ERA5 reanalysis for the years 2016 **(a, c, e)** and 2020 **(b, d, f)** as a function of latitude and altitude. The black dashed horizontal line indicates the tropopause from ERA5. Monthly mean zonal mean wind component, $u$ (m s$^{-1}$), from ERA5 is overlaid as solid gray contours (westerly) and dashed gray contours(easterly) lines.

depth in the lower stratosphere between the 2015–2016 and 2019–2020 QBO disruption events. During the 2015–2016 QBO disruption, the net wave forcing is stronger and broader in the lower stratosphere between the tropopause and the altitude of



about 25 $km$ than during the 2019–2020 QBO disruption (Fig 5a, b). Particularly, the wave breaking near the equatorward flanks of the subtropical jet known as a major BDC forcings region is narrower in 2020 than 2016. These differences in net

wave forcing are the main cause of a weaker advective BDC and its modulation by the QBO–induced secondary circulation in 2020 than in 2016, therefore, contributing to the anomalous lower stratospheric $H_2O$ and $O_3$ differences in addition to the significant Australian wildfire effect on lower stratospheric $H_2O$ mixing ratios. The wave forcing evolution during six months (January–to–June) after the QBO disruptions is consistent with the zonal mean differences in wave forcings, i.e. the time series of net forcing, planetary and gravity wave drag (Fig. S5).

In addition, we show the contribution of planetary (Fig 5c, d, and Fig. S5b) and gravity (Fig 5e, f and Fig. S5c) wave drag to better understand the role of each forcing in the anomalous circulation differences during both QBO disruption events. Beside the good agreement in the structure of planetary and gravity wave breaking, our analyses also show differences between the 2015–2016 and 2019–2020 disruption events in wave drag. The planetary and gravity wave drag indicates stronger anomalies in wave dissipation in the lower stratosphere between the tropopause and the altitude of about 25 $km$ during the 2015–2016

QBO disruption than during the 2019–2020 QBO disruption (Fig. 5c–f and Fig. S5b, c in the supplement). The anomalies in planetary wave dissipation associated with the 2015–2016 QBO disruption are stronger and extend from the tropics toward the subtropical jet between the tropopause and the altitude of about 25 $km$, while for the 2019–2020 disruption, these anomalies are smaller and confined to the tropics. These differences in the strength and depth of the anomalies are even larger in the gravity wave drag. During the 2015–2016 QBO disruption, gravity waves break in the entire lower stratosphere between the

tropopause and the altitude of about 25 $km$ with a maximum occurring near the upper flank of the subtropical jet, a key region for strengthening the shallow branch of the BDC (Shepherd and McLandress, 2011; Diallo et al., 2019, 2021) (Fig. 5e, f). The differences in the strength and depth of planetary and gravity wave breaking are clearly the main cause of observed differences in the anomalous upwelling strength of the BDC between the 2015–2016 and 2019–2020 QBO disruptions. This main cause is a combination of planetary wave dissipation in the tropics and particularly strong gravity wave breaking near the equatorward

flanks of the subtropical jet during the 2015–2016 QBO disruption as shown in the previous studies (Kang et al., 2020; Kang and Chun, 2021; Osprey et al., 2016). In summary, the strong planetary waves and gravity waves, which are likely related to ENSO and IOD, are responsible for the tropical upwelling of the BDC differences and its modulation by the QBO–induced secondary circulation, therefore, the negative lower stratospheric $H_2O$ and $O_3$ anomalies. Regardless of the net wave forcing in 2020, Australian wildfire led to less lower stratospheric dehydration due to the warmer stratosphere.

Note that during the 2015–2016 and 2019–2020 QBO disruptions, the surface conditions were different in terms of natural variability–induced convective activity. To trace back and link the potential source of convectively generated wave activities to regional differences, we finally analysed the monthly mean Outgoing Longwave Radiation (OLR) (Fig. 6 and Fig. S6 in the supplement). Clearly, there are regional differences in the occurrence of strong convective events between the 2015–2016 and 2019–2020 QBO disruptions. During the 2015–2016 QBO disruption, the tropical mean OLR anomalies reveal two active

convective regions, namely the East Indian Ocean associated with the negative IOD in 2016, and the Central Pacific Ocean associated with the 2015–2016 El Niño. However, during the 2019–2020 QBO disruption, the tropical mean OLR anomalies show only one strong active convective region that is the West Indian Ocean and East Africa associated with the strong 2019





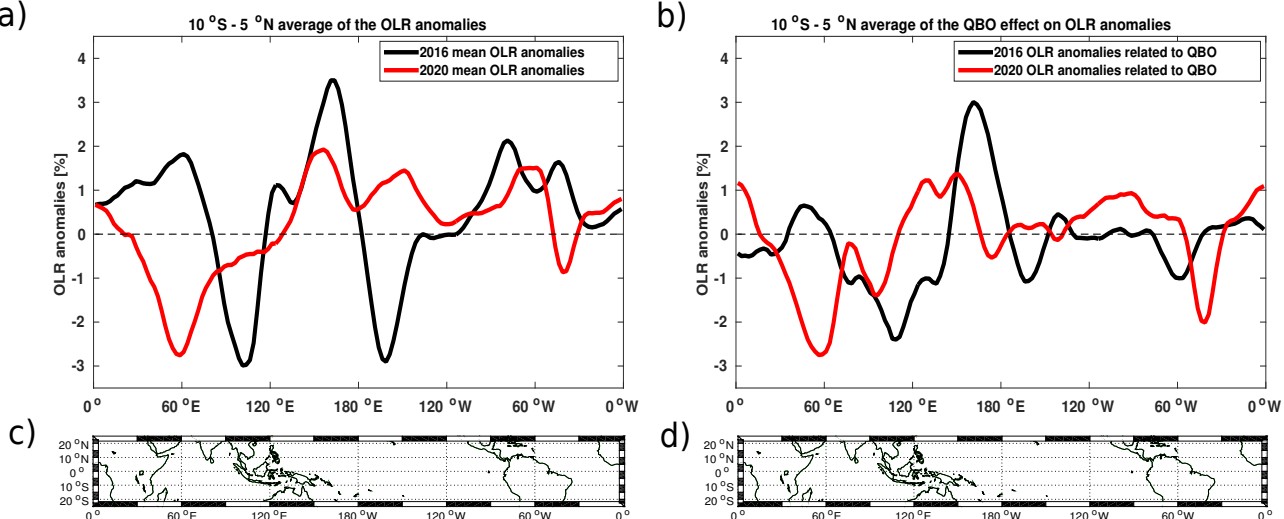

**Figure 6.** Longitudinal variations of the monthly mean Outgoing Longwave Radiation (OLR) anomalies **(a)** averaged between $20^o$ S–$20^o$ S together with the 2016 and 2020 QBO effect **(b)** associated with the convective activity derived from the multiple regression fit. The lowermost panels **(c, d)** shows the tropical region where the OLR timeseries are averaged.

IOD. Both QBO disruption effects related to OLR variations are linked to strong convective activity in the Indian Ocean, therefore, suggesting the importance role of this region may play in strong wave activities. This additional information related to the strength of convective activities in the Indian Ocean is of great interest for better understanding and relating the origin of the QBO disruption events and their strength based on regional forcings. This regional forcing and interplay of different modes of climate variability will be presented in further studies.

## 5   Summary and conclusions

Based on an established multiple regression method applied to Aura MLS observations, we found that both the 2015–2016 and 2019–2020 QBO disruptions induced similar structural changes in the lower stratospheric $H_2O$ and $O_3$ anomalies. Both QBO disruptions induced negative anomalies in $H_2O$ and $O_3$, few months after the sudden shift from the QBO westerly to QBO easterly wind shear reached the tropical tropopause. During the boreal winter of 2015–2016 (September 2015–March 2016), the alignment of the strong El Niño with the QBO westerly strongly moistened the lower stratosphere between the tropopause and the altitude of $25\,km$ (positive anomalies of more than 20 %). Analogously, the alignment of the weak El Niño with the strong QBO westerly and the impact of Australian wildfire smoke strongly moistened the lower stratosphere (positive anomalies of more than 20 %) during the boreal winter of 2019–2020 (September 2019–Jun 2020). The sudden shift from the QBO westerly to QBO easterly wind shear reversed the lower stratospheric moistening between the tropopause and an altitude of about $20\,km$, therefore leading to large negative $H_2O$ and $O_3$ anomalies by the end of summer 2016 and to small negative





H$_2$O and moderate negative O$_3$ anomalies in 2020. These decreases in H$_2$O and O$_3$ mixing ratios are due to a strengthening of
the tropical upwelling of the BDC and cooling tropical cold point temperatures as well as their modulations by the secondary
circulation induced by the QBO wind shear, consistent with the residual vertical velocity and temperature anomalies.

However, differences occur in the strength and depth of the QBO disruption–induced negative H$_2$O and O$_3$ anomalies in the
lower stratosphere between 2016 and 2020. We found that the impact of the 2019–2020 QBO disruption on lower stratospheric
H$_2$O and O$_3$ anomalies is smaller and shallower than the 2015–2016 disrupted QBO impact. The differences in the strength
and depth of the O$_3$ anomalies and its modulation by the QBO disruption events are due to discrepancies in the anomalous
tropical upwelling of the BDC, which was up to about 25 % larger in 2016 than in 2020. The analysis of the wave drag
shows that the differences in planetary wave breaking in the tropical lower stratosphere and the gravity wave breaking near the
equatorward upper flank of the subtropical jet are the main reasons of the differences in the anomalous tropical upwelling of
the BDC and secondary circulation between the year 2016 and the year 2020. The main differences in lower stratospheric H$_2$O
anomalies between the year 2016 and the year 2020 are due to discrepancies in the topical cold point temperatures. Despite of
the anomalous planetary waves and gravity wave activities, which are likely related to ENSO and IOD, the 2020 Australian
wildfire predominantly warmed the cold point temperature, therefore, leading to less dehydration of the lower stratosphere.

Finally, our results suggest that the interplay of QBO phases with a combination of ENSO and IOD events, and in particular
also wildfires and volcanic eruptions, will be crucial for the control of the lower stratospheric H$_2$O and O$_3$ budget in a changing
future climate. Especially, when increasing future warming will lead to trends in ENSO (Timmermann et al., 1999; Cai et al.,
2014) and IOD (Ihara et al., 2008) as projected by climate models, and a related potential increase in wildfire frequency
combined with a decreasing lower stratospheric QBO amplitude (Kawatani and Hamilton, 2013) are expected in future climate
projections. The interplay will change with strong El Niño/negative IOD and La Niña/strong positive IOD likely controlling
the lower stratospheric trace gas distributions and variability more strongly in a future changing climate. Clearly, both ENSO
and IOD impact on the tropopause height and tropical cold point temperature. Further analysis is needed using climate model
sensitivity simulations to pinpoint the impact of these future changes in lower stratospheric trace gases and the related radiative
feedback.

*Data availability.* MLS water vapor and ozone data were obtained from the Goddard Earth Sciences Data and Information Services Center
at es Center at doi.10.5067/Aura/MLS/DATA2508 and doi.10.5067/Aura/MLS/DATA2516, respectively. The aerosol optical depth data is
available through Khaykin et al. 2020. The ERA5 reanalysis are available at https://apps.ecmwf.int/data-catalogues/era5/?class=ea, last
access: 2nd February 2022, through Hersbach et al., 2020.

*Author contributions.* MD designed the study, conducted research, performed the calculation and the complete analysis of the impact of the
QBO disruptions as well as drafted the first manuscript. ME calculated the wave decomposition. FP, MH, ME, JU, SK and MR provided
helpful discussions and comments. MD edited the final draft with contributions from all co-authors for communication with the journal.



*Competing interests.*   The authors declare that they have no conflict of interest.

*Acknowledgements.*   Mohamadou Diallo research position is funded by the Deutsche Forschungsgemeinschaft (DFG) individual research grant number DI2618/1-1 and Institute of Energy and Climate Research, Stratosphere (IEK-7), Forschungszentrum in Jülich during which this work had been carried out. FP is funded by the Helmholtz Association under grant number VH-NG-1128 (Helmholtz Young Investigators Group A-SPECi). Manfred Ern was supported by the German Federal Ministry of Education and Research (Bundesministerium für Bildung

und Forschung, BMBF) project QUBICC, grant number 01LG1905C, as part of the Role of the Middle Atmosphere in Climate II (ROMIC-II) programme of BMBF. We gratefully acknowledge the Earth System Modelling Project (ESM) for funding this work by providing computing time on the ESM partition of the supercomputer JUWELS at the Jülich Supercomputing Centre (JSC). Moreover, we particularly thank the European Centre for Medium-Range Weather Forecasts for providing the ERA5 and ERA-Interim reanalysis data.



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
