# Peer review of "Stratospheric water vapour and ozone response to different QBO disruption events in 2016 and 2020"

_Atmospheric Chemistry and Physics, 2022_

## Author Comment (AC1)

**Answer to Reviewer #1 Comments on "Stratospheric water vapour and ozone response to different QBO disruption events in 2016 and 2020" by Mohamadou Diallo et al.**

Dear Editor-in-Chief, S. Fadnavis,

We are submitting our revised article titled "Stratospheric water vapour and ozone response to different QBO disruption events in 2016 and 2020". We thank the three Reviewers for their detailed and well thought-out comments, which helped to significantly improve the paper. We made substantial changes to the manuscript in order to thoroughly address the Reviewers' suggestions and comments. The main changes concern:

- Merging of the former Fig S3 with the Figure 1 and Figure 2 in the manuscript

- Moving the Fig S4 into the manuscript as suggested by Reviewer #1 & Reviewer #2 and the related discussion.

- Redone all figures to change wind contours, increase font size and to improve their quality.

- Rephrasing of certain paragraphs in order to clarify the manuscript.

With these changes, we are convinced that the paper is highly relevant for a wide-ranging journal like *Atmospheric Chemistry and Physics*. Please see below our answers point by point to all reviewers' comments and suggestions.

Reviewers comments are in bold, followed by our respective replies. Changes in the manuscript are in blue, allowing them to be tracked easily.
Kind regards,
Mohamadou Diallo (on behalf of the co-authors)

**Reviewer #1 (Comment on acp-2022-382):**

*Diallo et al. investigate the impact of the two QBO disruption events in 2016 and 2020 on water vapour and ozone using ERA-5 reanalyses and satellite observations from MLS. They find differences in the impact these disruption events had on atmospheric circulation and thus on the trace gas distribution of water vapour and ozone. This is a quite interesting study with interesting results. However, the writing could have been generally done a bit better and I have several suggestions for major revision before publication in ACP.*
*Generally, the whole study and writing is a bit too descriptive and though you state in the abstract that you "quantify" the impact it is done throughout the study in just a "qualitatively" manner. Don't understand me wrong, I do not need for everything numbers, but if there are too many phrases using terms like "weakly", "small", "large" it is quite difficult to get a feeling for how strong actually the impact is. I will provide more detailed feedback on this in the specific comments.*

**General Comments:**

*Usage of the term "2016" and "2020" and "2015-2016" and "2019-2020". In some occasions the whole period is used and in other only the second year of the period is used. I could not really see if there is a concept when you use which term, thus I would suggest to change to one way of writing consistently throughout the manuscript or explain when you use what.*
Thanks for the good suggestion. We have rephased and used only one concept of terminology.

*In all figures the font size should be increased. In the supplement this should be done for Figures 1-5.*
We have redone all figures and increased the font size and changed the wind contours.

*Use the Copernicus style: Units (km, hPa,. . . . . . .) are written in upright font.*
We have rephrased the units.

*Your results are based on the measurements from one satellite, namely MLS. I remember that there are significant differences in the QBO imprint on the trace gases between different satellites. How does that affect your results? Have you done a similar analysis using another satellite instrument? What where the differences?*

MLS water vapor and ozone products are the best available stratospheric trace gas measurements for the purpose here when considering both measurement uncertainty and sampling issues. Data from ACE-FTS, for example, has a much lower sampling frequency in the tropics, the key region for estimating QBO effects. Most of the satellite merged H2O and O3 products available (SWOOSH, BASICS,...) are actually using MLS. So we think using MLS data is the best we can do here.

***Some figures of the supplement as e.g Figure S5 should be moved to the manuscript since they are discussed in detail and seem thus to be not that unimportant. There are also some other figures in the supplement that also could be moved to the manuscript.***

Thanks for the good suggestion. Figure S5 is similar to the current Figure 6. We have rephrased the paragraphs (Pages 16–17) and moved former Figure S3 and S4 into the main text.

**Specific comments:**

1. ***P1, L6: "weakly decrease". Be more precise. Decrease by what? What exactly causes the decrease? The BDC transport?***

   We have rephrased the abstract (Page 1, line 1-20).

2. ***P1, L7: Here you talk about "circulation anomalies", but before you talked about changes in the trace gas distributions (their abundance). I would prefer a clear separation in the language between dynamical processes and their consecutive imprint on the trace gas distributions.***

   We have rephrased the abstract (Page 1, line 1-20).

3. ***P1, L17: Here a 1-2 sentence description what the BDC is should be added.***

   We have included a sentence defining the BDC and rephrased the paragraph (Page 2, line 23-43).

4. ***P2, L21: "Ozone is mainly produced in the middle stratosphere and is a good proxy for tropical upwelling". This is generally correct, but a too simple and not correctly understandable sentence for non-experts. I would suggest to rephrase this sentence and clearly state when and where is ozone produced, how is it transported and why can it be used as proxy for transport.***

   We have rephased the paragraph and included the suggestions (Page 2, line 23-43).

5. ***P2, L27: Introduce the abbreviation "QBO" and add a sentence describing what it is. You actually do that in the next paragraph. This paragraph should be moved higher up.***

   We prefer having the definition of the QBO in the paragraph reserved to the QBO and its impact on the trace gases. It makes more sense. The acronym "QBO" is now already explained in the first sentence of the abstract (Page 2, line 44-50).

6. ***P2, L29: Explain also shortly what dehydration is.***

   The dehydration is a common term meaning "the process of losing/removing water or moisture". We have included a definition (Page 2, line 40-41).

7. ***P2, L32: Shouldn't it read "e.g. water vapour and ozone". Doesn't this hold also for other trace gases?***

   Yes indeed the QBO impacts all stratospheric trace gases, including H2O and O3. We have rephrased the paragraph (Page 2, line 47-48).

8. ***P2, L31-36: As stated in my comment on P2, L27 this entire paragraph should be incorporated in the previous paragraph.***

   Thank you for the suggestion. We think it is better to have two separated paragraphs: one for the QBO and the other one for trace gases and their modulations for coherency. We have rephrased the two paragraphs. '

9. ***P2, L52: ....affect the radiative forcing it the Earth's climate system..." What does that mean? Are we (the society) affected by these disruptions? What are the changes or consequences we experience based on this disruptions? Or are these just interesting for scientists to better understand atmospheric circulation?***

   Water vapour and ozone are radiatively active trace gases, therefore, any change in their abundance in the UTLS will impact the the Earth's climate system, including surface temperatures. We have rephrased the sentence to be more explicit (Page 3, line 62-65).

10. ***P3, L68-69: "....high precision and lower systematic uncertainty...." Add some numbers. How high is the precision? Lower uncertainties than what? The former MLS version?***

We have rephrased the paragraph (Page 3, line 87-97)

11. ***P4, L112: Rephrase this sentence that either the references are incorporated in the text or so that these can be added in parentheses. As it is done know it is not correct***

We have rephrased the sentence (Page 2, line 53-56).

12. ***P4, L116: Differences in the disruptions. Are here references missing? Are you referring to previous studies or is this done in this study? Please clarify and revise text accordingly.***

We refer here to the differences in structure (strength and depth) shown in figure 1a. We have rephrased the sentence (Page 5, line 140-144).

13. ***P5, Figure 1 caption: How has the onset/offset be defined? When exactly did this happen. Can you provide year/month of the respective onsets and offsets?***

We have rephrased the captions of Figure 1, 2 and supplement.

14. ***P6, L148: What is the "tape recorder". A short explanation should be added.***

This is a common terminology first used in Mote et al. (1996) meaning "large-scale upward advection of the tropical stratospheric water vapor" as we explained it in the text. We have rephrased and added a reference (Page 7, line 179-181).

15. ***P7, Figure 2 description: I really have trouble follow your descriptions/explanations. This is really difficult to see from the figures. Could add some guidance for the eye in the figures, like arrows or boxes or any other shape or sign that marks the respective areas?***

Thank you for the suggestion. We have opted the recommendation of the Reviewer 2 by specifying the altitudes ranges and year periods. This option avoids overloading the figures. We have rephrased the paragraphs (Page 8).

16. ***P7, L181: I would suggest to rephrase this sentence. Dehydration refers only to H2O not to both H2O and O3. This sentence can be easily misunderstood.***

We have rephrased the sentence (Page 8, line 217-221).

17. ***P7-8: This is all bit too qualitative and difficult to see in the figures. Is there a possibility to quantify the changes?***

We have rephrased and added quantifications in the paragraphs (Page 8).

18. ***P8, L197: How do you exactly derive the "zonal mean impact"? What has been done/considered here? The difference of the zonal means?***

The impact of the QBO as well as the wildfire are both estimated using the regression analysis. We have rephrased the section 2 data and methodology (Page 4) and paragraph in page 8, line 190-195.

19. ***P8, L198: Which AOD data has been used?***

The AOD used is described in the method section 2.

20. ***P8, L198: How do you derive the impact? This becomes not clear.***

Please see the method section 2 and paragraph (Page 10, line 234-242).

21. ***P8, L216: "Also note the large variability...." Is this visible in the figures? Or do you mean these have been shown in other studies? If the latter is the case references should be added. If the former is the cause the text should be rephrased.***

We are discussing about the large ozone variability in the extratropics in Fig 3c. We have rephrased the sentence (Page 10, line 256-258).

22. ***P9, L226: "large" should be quantified or give more information on the differences than just "large".***

We have rephrased the sentence (Page 12, line 265-266).

23. ***P10, Figure 3 caption, 3rd line: The monthly mean mixing ratios you are referring here to; are these for the entire time period 2005-2020?***

The time period used here is 2005-2014 excluding the disrupted years. We have rephrased the sentence in the captions of Fig 3.

24. ***P10, Figure 3 caption: The sentence "The impact of the QBO ........." should be incorporated into the main text rather than in the figure caption.***

The terminology of "QBO disruption-induced changes in..." or "the impact of the QBO disruption on..." are equivalent. We have incorporated it through main text.

25. ***P10, Figure 10: The wind lines are difficult to see in detail. Thus, I would suggest to add a figure panel showing only the wind.***

The wind lines are the same shown in the Figure S1, therefore, we would like to avoid repeating the same figure into the main text. We have changed the contour of the zonal wind in all figures.

26. ***P11, L231ff: Can you quantify these differences?***

We have quantified these differences in O3 in the previous paragraph (Page 11, lines 250-255).

27. ***P11, L254: about 10% weaker? How do you derive this number?***

We have inferred the 10% estimate from the colorbar difference in current Fig. 1d and 2c. In 2016, w* anomaly reaches up to about 40% below 17 km while in 2020 w* anomaly barely reaches 25-30%. The difference leads to about 10%. We have rephrased the sentence (Pages 12-15, line 287-308).

28. ***P11, L259: Fig S4 and maybe some other figures should be rather moved to the main text. It is quite inconvenient to swap back and forth between the manuscript and the supplement.***

Thank you for the suggestion. We have moved the Fig. S4 to the main text and rephrased the paragraphs (Page 14).

29. ***P13, L280: Add a marker/box in the figure to better visualize this?***

We have opted to specify the altitude range as suggested by the Reviewer 2, therefore, avoiding to overload the plots. We have rephrased the sentences accordingly.

30. ***P16, L348: "large" and "small". Please quantify this.***

We have rephrased the paragraph and quantified the differences.

31. ***P17, L354: "smaller" and "shallower". Same here as for P16, L348.***

We have rephrased the sentence (Page 16, line 354-357).

32. ***P17, L358 and L360: This is really hard to see from the figures shown.***

Based on the suggestions from Reviewer 2, we have specified in the text the region of wave breaking, leading to BDC differences (Page 16, line 358-360).

33. ***Supplement, Figure 3: Although you can only show here a specific altitude range, these figures is much more helpful to see the difference. I would suggest to put this figure into the manuscript rather than in the supplement.***

Thank you for the good suggestion. We have moved the Figure S3 into the main text and dispatched it to Fig 2 and Fig. 3.

34. ***Supplement, Figure 5: Since this figure is discussed in detail in the manuscript it also should rather appear there than in the supplement.***

Thank you for the good suggestion. We have moved the Figure S3 and S4 into the main text and rephrased the paragraph related to Fig. S5.

**Technical Comments and Corrections:**

1. ***P1, L1-2: I would suggest to move "in the tropical stratosphere" to the first part of the sentence so that it reads: "The Quasi-biennial Oscillation (QBO) is a major mode of climate variability in the tropical stratosphere, with . . . . . . . . . . . . . . . ".***

   Thank you for the good suggestion. We have rephrased the sentence.

2. ***P1, L4: Writing it like this is rather misleading. I would suggest to rewrite the sentence as follows:. . . ..on the Brewer-Dobson circulation and respective distributions of water vapour and ozone, using. . . . . ...".***

   Thank you for the good suggestion. We have rephrased the sentence.

3. ***P1, L14: The line "Copyright statement: TEXT" is obsolete and can be deleted.***

   We have done it.

4. ***P1, L29: in the air parcels –¿ of the air parcels***

   We have rephrased the sentence.

5. ***P3, L70: Add "e.g". There are also other studies that document the quality of the MLS H2O data than the ones by Hegglin et al.***

   We have rephrased the sentence.

6. ***P3, L71: Here a capital "U" is used. Later to the wind with a small "u" is referred. This should be done consequently throughout the manuscript in one or the other way.***

   We have rephrased the terminology consistently.

7. ***P3, L81: Rephrase sentence as follows: "In the figures only the 2013-2020 period is shown to highlight the two QBO disruptions.***

   We have rephrased the sentence.

8. ***P4, L92: Introduce abbreviation "ENSO".***

   The ENSO abbreviation is already introduced in Page 3, line 77

9. ***P4, L95: Introduce abbreviation "AOD"***

   We have introduced the abbreviation "AOD" in Page 4, line 120.

10. ***P4, L99: In In –¿ In***

    We have rephrased the sentences (Page 4, line 126).

11. ***P4, L105: are → were***

    We have rephrased the sentence.

12. ***P5, Figure 1 caption: "U" or "u"?***

    We have rephrased the sentence.

13. ***P7, L158: 3-3 → O3***

    We have rephrased the sentence.

14. ***P7, L159: I am not entirely sure, but I would add "a", so that it reads "we performed a regression analyses"***

    We have added the article "a".

15. ***P7, L163: citet instead of citep***

    We have rephrased to citep.

16. ***P8, L186: disrution → disruption***

    We have rephrased the sentence.

17. ***P9, L221 and L225: JAS → July-August-September (or July-to-September)***

    We mean JAS → July-August-September. We have rephrased the sentence.

18. ***P11, L227: add "phase" or "winds" after "easterly"***

    We have added "winds" after "easterly.

19. ***P11, L229: JAS → July-August-September (or July-to-September)***

    We have rephrased the sentence.

20. ***P12, Figure 4 caption: "Tropical averaged of the deseasonalized mean" should be either changed to "Tropical averaged deseasonalized mean residual velocity" or to "Tropical averages of the deseasonalized mean residual velocity".***

    We have rephrased to "Tropical average of the deseasonalized mean residual velocity"

21. ***P13, L284: move "in the following" before "we finally" so that it reads "in the following we finally investigate......."***

    We have rephrased the sentence.

22. ***P14, Figure 5 caption: space between "(NetF)" and "(a,b)" and between "(contours)" and "easterly" missing.***

    We have rephrased the sentence.

23. ***Supplement: Check the figure captions. The units should be in upright font (same holds for the manuscript) and in several occasions the O3 in H2O is in italic instead of an upright font.***

    We have rephrased the sentence.

24. ***Supplement, Figure 3 caption: Add which line is the blue one and which is the red one.***

    We have rephrased the sentence.

---

## Author Comment (AC2)

**Answer to Reviewer #2 Comments on "Stratospheric water vapour and ozone response to different QBO disruption events in 2016 and 2020" by Mohamadou Diallo et al.**

Dear Editor-in-Chief, S. Fadnavis,

We are submitting our revised article titled "Stratospheric water vapour and ozone response to different QBO disruption events in 2016 and 2020". We thank the three Reviewers for their detailed and well thought-out comments, which helped to significantly improve the paper. We made substantial changes to the manuscript in order to thoroughly address the Reviewers' suggestions and comments. The main changes concern:

- Merging of the former Fig S3 with the Figure 1 and Figure 2 in the manuscript

- Moving the Fig S4 into the manuscript as suggested by Reviewer #1 & Reviewer #2 and the related discussion.

- Redone all figures to change wind contours, increase font size and to improve their quality.

- Rephrasing of certain paragraphs in order to clarify the manuscript.

With these changes, we are convinced that the paper is highly relevant for a wide-ranging journal like *Atmospheric Chemistry and Physics*. Please see below our answers point by point to all reviewers' comments and suggestions.

Reviewers comments are in bold, followed by our respective replies. Changes in the manuscript are in blue, allowing them to be tracked easily.
Kind regards,
Mohamadou Diallo (on behalf of the co-authors)

**Reviewer #2 (Comment on acp-2022-382):**

*Diallo et. al use ERA5 data and Aura Microwave Limb Sounder (MLS) satellite observations to quantify the impact of anomalous QBO events that occured in 2016/2017 and 2019/2020 on the Brewer-Dobson circulation, water vapour and ozone. They highlight the importance of understanding the reasons for disruptions in the QBO because of its impact on climate research within a changing climate by making use of multiple regression analyses to separate the impact of the QBO on the circulation, ozone and water vapour.*

**General Comments:**

*The paper is well written and well structured, but needs some more precision in some points to make concepts clearer (indicated below). The text is often very descriptive and it easily becomes tedious to read, but this is necessary in order to build the storyline and possible omissions have been applied where concepts for the first QBO disruption are similar to the second. The supplement only contains figures which are used extensively in the text. An effort should be made to include the really necessary images in the text and only put figures and add explanatory text in the supplement that is not necessary to understand the idea behind the paper. It does not help the reader to have to refer to the supplement to understand the main text.*
Thanks for these good suggestions. We have now moved figures and added corresponding explanatory text into the main text.

**Specific and technical comments:**

1. *Most comments I have are questions about understanding and precision. I noticed that you mix American (A) and British (B) spelling (center (A) vs centre (B), vapour (B) vs vapor (A)). Could you check for consistency?*
   We have chosen the British spelling and rephrased the text.

2. ***Line 1: What do you mean with "major mode of climate variability"? Do you want to speak about the disruption as a change to the QBO as a mode of climate variability? Climate change impacting this mode?***

   We are speaking about the QBO, which is a mode of natural climate variability, which modulates the year-to-year variability of the climate system.

3. ***Line 3: It sounds as if there was a fixed 28–month period for previous QBO periods when in fact it varied before. Maybe giving a range indicating in what the disruption consisted is better here.***

   The QBO has a predominant cycle of 28.3 months even though its mean period can vary between 28-months and 29 months. We have rephrased the sentence (Page 1, line 3 and Page 2, line 37).

4. ***Line 5: Better write "Both, water vapour and ozone in the lower stratosphere" instead of "Both lower stratospheric trace gases"***

   Thanks for this good suggestion. We have rephrased the sentence (Page 1, line 5-6).

5. ***Line 7: Do you mean "anomalous circulation response" instead of "circulation anomalous response"?***

   Thanks. We have rephrased the sentence (Page 1, line 8).

6. ***Line 11: Do you mean "hiding/obscuring/concealing" instead of "hidding"?***

   Thank you. Yes that's what we mean. We have rephrased the sentence (Page 1, line 12).

7. ***Lines 21/22: The two following sentences essentially say the same thing: "Ozone is mainly produced in the middle stratosphere and is a good proxy of the tropical upwelling. In addition, ozone variability in the tropical lower stratosphere is affected by variability in tropical upwelling of the BDC." Please revise.***

   We have rephrased the sentences (Page 2, line 23-26).

8. ***Lines 24/27: Do you mean "natural climate variability, including the QBO" or "modes of climate variability, such as the QBO"? The term "natural mode of climate variability" is confusing.***

   We have rephrased the wording (Page 2, line 28-29).

9. ***Lines 33/37: "oscillation between tropical westerly and easterly zonal wind shears" Do you mean "oscillation of the zonal wind"? The easterly and westerly shear zones descend differently.***

   We have rephrased the sentence (Page 2, line 37, 43 ).

10. ***Line 34: "QBO phases" You have not defined what you mean with phase, here. Looking at a vertical profile the QBO has easterly and westerly phases at different altitudes, so for the Brewer-Dobson circulation one might argue that there is on average no influence.***

    Yes indeed the QBO has easterly and westerly phases at different altitudes, which alternate every 28-29 months. In addition, the easterly and westerly shear zones descend differently therefore it affects BDC as the phases do not have the same length at different altitude as well. The QBO easterly are oftentime shorter that the QBO westerly. We have rephrased the sentence (Page 2, line 39).

11. ***Line 38: It was not the "anomalous QBO westerlies" but the "QBO westerlies" that got disrupted (by an anomaly).***

    We have rephrased the sentence (Page 2, line 43).

12. ***Line 44: I would write "climate change" instead of "climate changes"***

    We have rephrased the sentence (Page 2, line 49).

13. ***Line 46: Osprey et al 2016 do not mention CMIP6?!***

    We have rephrased the sentence.

14. ***Lines 54/55: This sentence appears not to be grammatically correct because verb and substantive have similar forms. It is better to reformulate the sentence, e.g: "Here we use satellite observations to quantify the similarities and differences in the strength and depth of perturbed/disrupted QBO effects in 2015-2016 and 2019-2020 on water vapor and ozone in the lower stratosphere. "***

    Thanks for this very good suggestion. We have rephrased the sentence (Page 3, line 60-62).

15. *Lines 60 to 63: Please reform the sentence to make your statement more lucid. The same issue of verb and substantive confusion might occur especially for the non English native reader. You probably mean something like: "Finally, we discuss the main reasons for the anomalous differences in BDC and UTLS composition between the 2015-2016 and 2019-2020 perturbed QBO effects associated with planetary and gravity wave dissipation, which are likely caused by the anomalous surface conditions associated with the strong El Niño Southern Oscillation (ENSO) in 2015-2016 and the strong Indian Ocean Dipole (IOD) in 2019-2020. "*

    We have rephrased the sentences (Page 3, line 67-70).

16. *Lines 63/64: Maybe better: "We also discuss the differences between 2016 and 2020 in terms of the particularly warm stratosphere in the context of Australian wildfire smoke in 2020."*

    We have rephrased the sentence (Page 3, line 71).

17. *Line 69: "lower systematic uncertainty" lower with respect to what?*

    We have rephrased the sentence (Page 3, line 81).

18. *Line 70: It is better to explain "multi-instrument mean"? From the text it is not obviously clear what you mean without reading Hegglin et al., 2013, 2021*

    We have clarified the sentence (Page 3, line 81-84).

19. *Line 71: I presume that the ERA5 data used is also 2005-2020?*

    We have added the time period for ERA5 (Page 3, line 86-87).

20. *Line 79: "...impact on these monthly..." should be "...impact on the monthly..." or "...impact on the MLS monthly..."*

    Yes, that is correct. We have rephrased the sentence (Page 3, line 81-84).

21. *Line 81: "To highlight the two QBO disruptions, figures only show the 2013–2020 period." Do you mean "To highlight the impact of the two QBO disruptions, figures only show the shorter 2013–2020 period."?*

    Thank you for this very good suggestion. We have rephrased the sentence (Page 3, line 60-62).

22. *Is the water vapour and the ozone from ERA5 much different from the MLS data? Because it would actually be really good to see the impact of the QBO on water vapour and ozone from 2005 to 2013 to visually see what "normally" happens, maybe as supplement.*

    Thank you for this very good suggestion. Our Konopka et al, 2022 GRL have comopared the trends between MLS and ERA5 and found very good agreement. Assessing the differences of the QBO impact on water vapour and the ozone from ERA5 and the MLS data is out of scope of the paper. Therefore, we leave that for a future study.

23. *Line 89: In this context (mathematical, technical) I prefer "indices" over "indexes". See for example $https://books.google.com/ngrams/graph?content=climate+indexese=true$*

    Thank you for this very good suggestion. We have rephrased the sentence (Page 3, line 60-62).

24. *Line 90: Please define "tropical" here. (for example averages over 5S to 5N)*

    We have rephrased the sentence (Page 4, line 105-106.

25. *Line 93: Please define "AOD"*

    We have rephrased the sentence (Page 4, line 109-110).

26. *Line 95: "The solar forcing is neglected because our data set is relatively short." You have a data set comprising 16 years. That means that you have more than one solar cycle. I don't think that this is short, especially not because a linear trend does not take into account the end of solar cycle 23 and solar cycle 24. Have you checked whether it matters? Is it too much work to include the solar forcing?*

    Thank you for the suggestion. One and half solar cycle is too short to include in the regression analysis because of the long time-lag needed for decadal variability indices.

27. *Line 99: There is an "In" too many at the beginning of the sentence.*

    We have rephrased the sentence (Page 4, line 115).

28. ***Line 99: "unexpected tropical QBO easterlies (negative QBOi) developed in the center" Here "in the center" is not clear, what is the center of the QBO? Is there a center? Over which altitude does the QBO exist? There are many questions that you raise by using "center" here. Maybe it is better to say at 22 km). Furthermore, it seems that the disruption already starts earlier that where you indicate at an altitude of 32 km. As the wind shear seems to shift downwards the disruption of the QBO may have been started earlier. By emphasizing your study on the disruption in the altitude range 15 to 24 km you might oversee something? Or are there two independent disruptions, one aloft and one starting at 22 km? The interesting thing is why at 26 km the westerlies persist.***

We agreed about the terminology. Regarding the disruption, it did not start early at least according to the literature. The appearing of the QBO easterly in the lower stratosphere is defined as the disruption actually, which upward shifted the descending QBO westerly above 26 km. We have rephrased the sentence (Page 4, line 116).

29. ***Line 112: The references should not be in brackets.***

We have rephrased the sentence (Page 4, line 138-140).

30. ***Line 114: Here you use "center" again. Better indicate the altitude range or 'center of the image'.***

We have rephrased the sentence (Page 5, line 138-144).

31. ***Line 119: You say that the disruption is visible in the water vapour ozone plots. This is difficult to judge if you only show 2013 to 2021. In the water vapour plot it even looks as if water vapour shows a strange behaviour before the onset of the disruption in 2016 that you indicate. Which would point towards a previous event maybe the onset of easterlies at 32 km at the beginning of the year. Tropical ozone anomalies are closely related to temperature anomalies show the QBO disruption, as you say, and are therefore to be expected.***

The 2013-2015 period is associated with the normal QBO, where we can see the inprint of the descending QBO easterly and westerly on the trace gas anomalies. It is well admitted in the community that the onset of easterlies is at about 26 km. At 32 km, it is onset of next QBO easterly, which was supposed to propagate downward after the QBO wersterly. In addition, we have previously published a paper regarding the 2015-2016 QBO disruption and the strong El Nino, where actually the entire time series were shown from 2005 to 2017 (Diallo et al 2018).

32. ***Line 125: "is the most effective" should be "is most effective". This sentence is not clear. Do you mean that the disruption impacts tropical upwelling via its impact on tropical upwelling? Do you mean the water vapour anomaly minimum at 17 km and the ozone anomaly minimum at 16 km the end of the disruption that you indicated? What do you mean by "when the signal reaches"? Why do you think that this is due to the disruption impact on tropical upwelling (if this is what you meant to say)?***

Thanks for this comment. We mean that the impact of the QBO disruption on H2O and O3 is stronger when the anomalous QBO easterlies (signal) reach the cold point temperatures because of its impact on tropical upwelling. We have rephrased the sentence (Page 7, line 154-156).

33. ***Line 127: You refer to the wind at 30 hPa but this is not shown in the figure. Please add the pressure altitudes to the figure or indicate altitudes in km (with pressure in brackets) whenever you mention pressure altitude in the text. Also indicate where this event happens. Do you mean at 26 km between the two vertical lines indicated, i.e. the uninterrupted westerlies during the disruption?***

Thank you for the good suggestion. We have rephrased the sentence and indicated the altitudes in km (25km) (Page 7,line 157-158).

34. ***Line 129: I would not speak about an upward shift of the westerlies. Westerlies are rather maintained longer and reestablish at the top moving downwards.***

We have rephrased the sentence (Page 7, line 160).

35. ***Line 133: "substantially increased H2O mixing ratios and decreased O3 mixing ratios" do you mean "coincided with an increase of H2O mixing ratio anomalies and a decrease of O3 mixing ratios anomalies"? This is true for water vapour anomalies but for the ozone anomalies there is an earlier onset with a slight increase and it does not reach as high up as the water vapour (25 km).***

This a conclusion of our previous findings (Diallo et al 2018) that we report here. As we are talking about the anomalies of both trace gases, therefore, we defined an altitude range in which their changes are happening "between the tropopause and the altitude of 25 km". Of course the maximum $H_2O$ and $O_3$ anomalies are below 25km but they are still in the defined range (16–25km). This small increase in ozone mixing ratio remains below 15 km, therefore, located below the tropopause level, which is of about 16 km in the tropics. We have rephrased the sentence (Page 7, line 164-165).

36. ***Line 134/135: "sudden occurrence of the QBO disruption". An interesting question here is if the disruption was not caused by this, and the ozone and water vapour response is just due to the thermodynamic balance.***

Yes indeed it is an interesting question. Note that the disruption is caused by wave propagation toward tropics, therefore, prior to the water vapor and ozone response even though these trace gas responses may feedback to the anomalous circulation (Page 7, line 166-167).

37. ***Line 138: the spelling of structural is wrong here (strutural).***

We have rephrased the spelling (Page 7, line 168).

38. ***Line 145: "compare" should be "compared" (or compares but then there should be an "and" before "suggests".***

We have rephrased the sentence (Page 7, line 177).

39. ***Line 147: Figure S3 is mentioned before S2 please switch the order of the figures.***

We have adjusted the text after including the Fig. S3 in the main text (Page 7, line 178).

40. ***Line 158: Spelling: "33 anomalies" should be "O3 anomalies"***

We have rephrased the sentences (Page 8, line 192).

41. ***Line 160: "The difference... gives the QBO-induced impact." Is it not only "the linear part of the QBO-induced impact"?***

The regression model contains a time-lag, allowing to capture the impact of QBO on these trace gases.

42. ***Line. 170: I don't really see this. I presume that you refer to the region between the two vertical lines. There ist seems the other way round? Or for water vapour it is negative when the QBOi increases (from negative to positive).***

When the QBO increases from negative to positive e.g. from westerly to easterly, the $H_2O$ and $O_3$ vary from positive anomalies to negative anomalies because of the impact of the QBO on tropical upwelling. The QBO easterly is associated with negative cold point anomalies (Plumb and Bell, 1982) and enhances upwelling, leading to negative ozone and water vapor anomalies. In addition, the water vapor response is always time-lagged by about a few months because of its tropospheric origin. The QBO westerly does the opposite. This is clearly visible for all years. It is also the case for 2020, even though this is a special year because of the Australian wildfire in 2020.

43. ***Line 171: Here I would only mention that the ozone anomaly changes follow closely the disruption in the zonal wind. I would not speak about suddenness.***

We have rephrased the sentence (Page 8, line 205).

44. ***Line 173: Here it would help to have the months as minor tick marks. I can see the alignment with ozone but for water vapour this is only speculation and is not true further up, i.e it might only be true for one altitude level.***

The water vapor response also is delayed by about a few months. Adding the months tick marks will overload the xlabel of the figures this is the reason we decided to not add them. We have rephrased the sentence (Page 8, line 205-208).

45. ***Line 179: 40 hPa please also give the altitude.***

We have rephrased the sentence (Page 8, line 215).

46. ***Line 179/180: The response is different for the ozone and water vapour anomalies. For ozone the altitude range and the temporal extend true but for water vapour it is rather from mid 2016.***

We have rephrased the sentence (Page 8, line 215-217).

47. *Line 182: "anomalously cold point temperatures" should be "anomalously low cold point temperatures"*

We have rephrased the sentence (Page 8, line 219)

48. *Line 183: Why is there enhanced tropical upwelling?*

There is enhanced tropical upwelling because of the presence of QBO easterly in the lower stratosphere, which is associated with the secondary circulation modualing the BDC (Plumb and Bell, 1982), which modulates the upwelling.

49. *Line 184: "indueced" should be "induced"*

We have rephrased the sentence (Page 8, line 221).

50. *Line 186/187: I don't understand: 1st statement: disruption induced O3 anomalies are small between 2016 and 2020; 2nd statement: disruption induced O3 differences are large between 2016 and 2020. I am not sure what you want to say here. Could you reformulate this sentence please to make it clearer?*

We have rephrased the sentence in order to enhance clarity.

51. *Line 192: What do you mean with "early" in "...we note that the early QBO westerly..."*

We mean the QBO westerly pre-disruption. We have rephrased the sentence.

52. *Line 197: "zonal mean impact" what do you mean with "impact"?*

By "impact" we mean the QBO–induced impact/effect on H2O and O3 shown in zonal mean plot.

53. *Line 201: Are those responses due to the disruption or due to the QBO in general? I.e. Do the responses just follow the stratospheric wind regime no matter if there is a disruption or not?*

Figure 3 shows the H2O and O3 responses due to the QBO disruptions. Yes, their responses also follow the stratospheric wind regimes such that negative anomalies are associated with QBO easterlies and positive anomalies with QBO westerlies.

54. *Line 204: "below the altitude of 20 km": at about 20 km there is the maximum hydration?! It is rather 18 km? You could add minor tick marks to the plot to see clearly where the sign changes.*

We have rephrased the sentence (Page 8, line 244).

55. *Line 204ff: How can you be sure?*

The H2O and O3 response in the tropics is mainly due to dynamical mechanism. We have discussed that in Sect. 4.2, page 12.

56. *Line 209f: "but stronger in winter hemisphere" Do you mean "but more in the winter hemisphere" or "but is stronger in the winter hemisphere"?*

We have rephrased the sentence (Page 10, line 247-250).

57. *Line 212: "consistent" Do you mean "correspond"?*

We mean "agrees/matches/consistent with" We have rephrased the sentence (Page 10, line 252).

58. *Line 226: "particularly large" please indicate what figures you are referring to.*

This is true for all figures. We have added that we are refering to (Fig 2, 3, S2 and S3).

59. *Line 234: "induced secondary circulation are weaker" should I not see this in figure S4a,b?*

The weaker secondary circulation is visible in Fig. 3 and 5c, d, g, h. We have added the reference (Page 12, line 276).

60. *Line 260: It is very difficult to follow your argument here, especially because the tropospheric w\* differences look very strange. In your argument you don't mention altitude ranges so which makes it difficult to follow.*

Here were are refering to Fig 4a, b and Fig. 4S. The tropospheric w\* does not matter here, but the tropical upwelling of the BDC changed between the tropoapuse ( 16km) and the altitude of 23 km is what matters for stratosphere (Page 10, line 247-250).

61. ***Line 264: "(Fig. 4c in the supplement)" does not show cold point temperatures. What do you mean with substantially in "substantially negative"? Maybe better "strongly negative"?***

We have rephrased the sentence (Page 15, line 305-306).

62. ***Line 266 Figure 4 which you refer to does not show water or ozone? Neither does Figure S4?! Please correct.***

We were refering to temperature and w*. We have rephrased the sentence (Page 10, line 305-308).

63. ***Line 279 "hidding" should be "hiding"***

We have rephrased the sentence (Page 15, line 320).

64. ***Line 297 "net wave forcing is stronger and broader" do you mean the red region between 20 and 25 km altitude?***

No we mean the blue region, as it is negative EP-flux divergence which is associated with wave breaking. The red regions mean less waves breaking. It is not straightforward to put an altitude as the region change with time as well (see the supplement). Indicating always the altitude range maybe misleading as the area of wave breaking varies as a function of latitude as well (Fig. 4). We have rephrased the sentence (Page 17, line 337-339).

65. ***Line 298 "wave breaking near the equatorward" do you mean the blue region between 18 and 22 km between latitudes of about -20S and +17N?***

The position of the subtropical jet is well known in the literature to be centred at 30N/S. The equatorward upper flank of the subtropical jet is the region between 10-30S/N above the tropopause level. We have rephrased the sentence (Page 17, line 334).

66. ***Line 303ff: "The wave forcing evolution..." what do you mean? Adding the components gives you the total? I guess I misunderstand this sentence?***

We mean "the variation in time of wave forcings". We have rephrased the sentence.

67. ***Line 322: What do you mean with "for the tropical upwelling of the BDC differences"?***

We mean "the differences in the tropical upwelling". We have rephrased the sentence.

68. ***Line 341: "few months after the sudden shift from the QBO westerly to QBO easterly wind shear reached the tropical tropopause. " For ozone it seems to have happened already earlier!***

we have rephrased the sentence.

69. ***Line 349ff: How can you be sure of this statement? ("strengthening of the tropical upwelling of the BDC")***

That is what we have shown with the analysis of the tropical upwellling of the BDC (w*).

70. ***Line 362 "warmed the cold point temperature" should be "raised the cold point temperature"***

We have rephrased the sentence.

71. ***In the Figure captions you repeat the definitions for a, b, c etc very often. It would be good to remove the repetitions.***

We think it is better to keep these few repetition in order to have a clearer description in the captions.

72. ***In the Figure captions you repeat the definitions for a, b, c etc very often. It would be good to remove the repetitions. Figure 1: I would prefer omitting the first (a) and putting the ERA in the description under the second (a).***

We have removed the first (a).

73. ***"50hPa" it would be good to have an approximate altitude at 50 hPa to see where we are in the panels (a) to (c).***

We have added the approximated altitude and rephrased the text.

74. ***The QBOi line looks weird because of the disruptions. Also here it would be good to see how regular it looked before. Please add months as minor tick marks.***

The 2013-2014 is a regular QBO years, showing the QBO effect. We have decided to not use month tick marks because of esthetical reasons.

75. ***Figure 4: The labeling seems wrong in the first sentence. What you call b should be c and vice versa. What you call "impact" I presume comes from the multiple regression analysis?***

    We have rephrased the sentence. By "impact" we mean the quantified effect of QBO using the regression model.

76. ***Figure S1: The altitudes in the text refer to pressure altitudes whereas here are only altitudes in km. Please make consistent. The whole year average for Figures a) and b) depends a lot on the phase of the QBO for that year.***

    We have given the corresponding altitude–pressure levels in the main text.

77. ***It appears that Figure S3 is mentioned before Figure S2 in the text, please reverse order.***

    We have rephrased the sentence.

78. ***Figure S2: There is a mistake: instead of 2016 (a, c) and 2020 (b, \*c\*) it should be 2016 (a, c) and 2020 (b, \*d\*).***

    We have rephrased it.

---

## Author Comment (AC3)

**Answer to Reviewer #3 Comments on "Stratospheric water vapour and ozone response to different QBO disruption events in 2016 and 2020" by Mohamadou Diallo et al.**

Dear Editor-in-Chief, S. Fadnavis,

We are submitting our revised article titled "Stratospheric water vapour and ozone response to different QBO disruption events in 2016 and 2020". We thank the three Reviewers for their detailed and well thought-out comments, which helped to significantly improve the paper. We made substantial changes to the manuscript in order to thoroughly address the Reviewers' suggestions and comments. The main changes concern:

- Merging of the former Fig S3 with the Figure 1 and Figure 2 in the manuscript

- Moving the Fig S4 into the manuscript as suggested by Reviewer #1 & Reviewer #2 and the related discussion.

- Redone all figures to change wind contours, increase font size and to improve their quality.

- Rephrasing of certain paragraphs in order to clarify the manuscript.

With these changes, we are convinced that the paper is highly relevant for a wide-ranging journal like *Atmospheric Chemistry and Physics*. Please see below our answers point by point to all reviewers' comments and suggestions.

Reviewers comments are in bold, followed by our respective replies. Changes in the manuscript are in blue, allowing them to be tracked easily.
Kind regards,
Mohamadou Diallo (on behalf of the co-authors)

**Reviewer #3 (Comment on acp-2022-382):**

***Diallo et al. present results from a multiple regression analysis of water vapor and ozone data from MLS and wind and temperature fields from ERA50 spanning 2013–2020 to delineate the impacts of QBO from other natural variations (e.g. El Niño) and from time-varying forcings (i.e. aerosol optical depth). They find distinct planetary wave forcing patterns corresponding to each of the two QBO disruptions, and ascribe the anomalously moist lower stratosphere during the 2020 disruption to Australian wildfires. I have concerns about the robustness of the multiple linear regression and the statistical methods used on such a short time series. I think the paper itself could use editing for flow, clarity, and language. The figures show much more information than is actually discussed, making it a chore for the reader to discern the meaningful results.***
Thanks for this critical, but very constructive general comment. We did a substantial rewriting of several parts in order to enhance clarity and language (as also explained in more detail in the responses to reviewers 1 and 2). Regarding the regression there is likely a misunderstanding, as we don't apply the regression just to the few years shown, but to the entire 2005-2020 period (right??). Hence, we think that the regression results are robust. More specific responses are given to the more specific comments below.

**General Comments:**

***Does the paper address relevant scientific questions within the scope of ACP?***
*Yes. The objective of this paper is important and germane to ACP.*

***Does the paper present novel concepts, ideas, tools, or data?***
*Yes. The multiple regression analysis including both the recent QBO disruptions is timely and useful.*

***Are substantial conclusions reached?***
*I think so; the location of the eddy forcing is helpful to know, and the attribution of moist UTLS during the 2020 disruption to the Australian wildfires – an impact that has been discussed in the community – is important to show.*

*Are the scientific methods and assumptions valid and clearly outlined?*
*I don't believe they are clearly outlined. I think the multiple regression analysis on 8 years of data – dealing with oscillations that are subseasonal-to-nearly-interdecadal in scope – may be over-determined. Furthermore, while the authors state that a t-test is used to test for statistical significance, they do not specify the parameters used (e.g.,the effective degrees of freedom, which I believe to be important in a time series that is temporally over-sampled with respect to the QBO, ENSO, etc.).*

Thanks for these remarks. This comment is very likely a misunstanding of the Reviewer. In Page 4, lines 105-110, we have described that the regression analysis that is applied for the 2005-2020 period e.g. 16 years of data, therefore, there is not issues of temporally over-sampling. In the figures, only the 2013-2020 period is shown to highlight the impact of the two QBO disruption events. The relative changes shown here give us the relance of the QBO effect on these trace gases.

*Are the results sufficient to support the interpretations and conclusions?*
*I don't believe so. Again, I think – considering the shortness of the MLS record – that the multiple regression analysis used here is likely to appear more significant than it is. Also, there is an implicit assumption in the test that the QBO disruptions are well described by a one-dimensional QBO index time series. In other words, that the impacts of the QBO are the same as the impacts of any other transition between QBO phases with a similar index. If this is indeed the case, the QBO disruptions may not be so interesting. If this is not the case, then the interesting impacts of QBO disruptions seem more likely to appear in the residual term (epsilon), but this is not examined in detail.*

As explained above, the regression covers the entire 15 years of MLS measurements 2005-2020 (actually 16 years). We agree that it would be better and increase robustness if we'd use an even longer time series, but unfortunately the MLS observations are restricted to that period. We also agree that the linear regression method is not capable of correctly separating non-linear effects. Nevertheless, our MLR results with an automated lag-time show a rather consistent picture between composition and dynamical changes, which in our opinion shows that the MLR method provides meaningful results here.

*Is the description of experiments and calculations sufficiently complete and precise to allow their reproduction by fellow scientists (traceability of results)?*
*I don't believe so, for reasons stated above. Also, Figure 3E is said to capture the aerosol optical depth (AOD) impacts of the Australian wildfires, but since there are only two disruptions, of differing character, it seems possible that some of the impact of the QBO disruption itself may be wrapped up in the AOD term, in addition to the residual term.*

Again, I'd suggest to argue with the meaningfulness of our results, the consistency between composition and dynamics variability. As far as we know there was not a wildfire during the 2015-2016 disruption. Figure 3e shows only the AOD impact. In addition, the regression is suitable and well established and has been applied in a previous aerosol impact study (Diallo et al. 2017, GRL).

*Do the authors give proper credit to related work and clearly indicate their own new/original contribution?*
*I believe so. The authors cite a good variety of work, including very recent work, and the authors' own related contributions. It seems strange to me that Taguchi (2010) is being cited to back up the statement that there is not yet a clear understanding of how QBO disruptions are linked to anomalous sea surface temperatures, since QBO disruptions had never occurred when that paper was written.*

We have rephrased the sentence.

*Does the title clearly reflect the contents of the paper?*
*Yes.*

*Does the abstract provide a concise and complete summary?*
*Yes*

*Is the overall presentation well structured and clear?*
*The paper could use some work in this regard. Paragraphs are rambling, having at times several different subjects. There are no subsections to aid quick navigation of the paper.*

We have rephrased many paragraph in order to improve clarity.

*Is the language fluent and precise?*
*The paper needs editing to be publication ready. Like the paragraphs, the sentences are long and often have weak structure. Definite/indefinite articles are often missing, singular/plural disagreement is prominent. Many other phrases (e.g. "the quasi-periodic QBO cycle of about 28–month period" on line 37) are just a bit awkward and could use careful reading by a native English speaker.*

As said above, we did a substantial effort of rewriting to improve language and clarity. We think that the text is substantially improved now.

*Are mathematical formulae, symbols, abbreviations, and units correctly defined and used?*
*Yes.*

*Should any parts of the paper (text, formulae, figures, tables) be clarified, reduced, combined, or eliminated?*
*I think many of the figures show 2D anomalies, but only certain levels or time periods are described. Perhaps reducing the dimensionality of the figures would make the authors' message clearer. Figure production is rough, with uniformly small labeling for ticks labels, and figure panel titles. The figure lettering is added after the fact, is large, and is removed from the figure panel title. Some figure labels are missing. All color scales are the same.*

We have increased the ticks labels and figure panel titles also suggested by the reviewers 1 and 2. We think that the visibility and clarity of figures is much improved now.

*Is the amount and quality of supplementary material appropriate?*
*I think the supplementary figures are of similarly rough quality to the figures in the main text. The figures I would be most interested in (some of the other regression terms) are not shown.*

The main focus of the paper is on the differences in BDC and the impact of the QBO disruption on H2O and O3.

**A few small edits/suggestions follow:**

1. *Line 7: "circulation anomalous responses" needs re-wording.*

   We have rephrased the sentence.

2. *Lines 9–11: this sentence needs to be split or otherwise clarified.*

   We have rephrased the sentence.

3. *Line 31: "Considered as a..." could just be "Considered a..."*

   I am not a native speaker, but starting the sentence with just "Considered a ..." does not sound odd to me, and could be used.

4. *Line 37: "The quasi-periodic QBO cycle of about 28–month period" needs rewording.*

   We have rephrased the sentence following the suggestion of Reviewer 2.

5. *Line 44: "study" should be "studies".*

   We have rephrased the sentence by removing the citation of Osprey et al (2016) as the study did not use CMIP6.

6. *Line 50: comma needed after amplitude.*

   We leave it as it is.

7. *Line 74: "planetary (PWD) and gravity (GWD) wave drag" would be better as "planetary wave drag (PWD) and gravity wave drag (GWD)." It's just two words more, but much easier to read.*

   We have rephrased the sentence.

8. *Line 99: "In in"*

   We have rephrased the sentence.

9. *Lines 104–106: Sentence beginning with "Both" is then used to contrast the two QBO events. Re-word.*

   We have rephrased the paragraph.

10. *Lines 161–163: This sentence is confusing. I keep reading "fits" as a verb, and it breaks everything. Make it clear.*

    We have rephrased the sentence.

11. *Lines 282–285: The last sentence of the previous paragraph and the first sentence of the subsequent paragraph are a bit redundant—they both serve to introduce the topic of gravity wave drag.*

    We think that the first sentence is for transition and the second for actually underlying what we do. We keep the sentences for clarity.

12. **Line 327: The word "finally" is repeated. Check how many times it's used. Use it once.**

We have checked the use of the word "finally". We used it only 5 times in the whole manuscript, and in different context. Therefore, we keep the wording as is.